# Performance Evaluation of the New High-Sensitivity Cardiac Troponin I Immunoassay on CL-2600i Mindray Analyzer

**DOI:** 10.3390/diagnostics15081031

**Published:** 2025-04-18

**Authors:** Flaminia Tomassetti, Denise Fiorelli, Edoardo Cappa, Alfredo Giovannelli, Martina Pelagalli, Silvia Velocci, Eleonora Nicolai, Marilena Minieri, Marco Alfonso Perrone, Sergio Bernardini, Massimo Pieri

**Affiliations:** 1Department of Experimental Medicine, University of Rome Tor Vergata, Via Montpellier 1, 00133 Rome, Italy; flaminia.tomassetti@students.uniroma2.eu (F.T.); fiorellidenise@gmail.com (D.F.); edoardocappa@hotmail.com (E.C.); alfredo.giovannelli@gmail.com (A.G.); velocci.silvia@gmail.com (S.V.); bernards@uniroma2.it (S.B.); massimo.pieri@uniroma2.it (M.P.); 2Department of Laboratory Medicine, Tor Vergata University Hospital, Viale Oxford 81, 00133 Rome, Italy; pelagallimartina90@gmail.com; 3Department of Biomedicine and Prevention, University of Rome Tor Vergata, 00133 Rome, Italy; 4Department of Industrial Engineering, University of Rome Tor Vergata, 00133 Rome, Italy; 5Departmental Faculty of Medicine, Unicamillus-Saint Camillus International University of Health and Medical Sciences, Via di Sant’Alessandro 8, 00131 Rome, Italy; eleonora.nicolai@unicamillus.org; 6Department of Clinical Sciences and Translational Medicine, University of Rome Tor Vergata, Via Montpellier 1, 00133 Rome, Italy; marco.perrone@uniroma2.it

**Keywords:** hs-cTn, chemiluminescence assay, troponin, method validation

## Abstract

**Background**: International guidelines recommend the use of high-sensitivity cardiac troponin (hs-cTn) I and T methods for the detection of myocardial injury as a pre-requisite for the diagnosis of acute myocardial infarction (AMI) in patients admitted to the emergency department. Recently, Mindray (Mindray Bio-Medical Electronics Co., Ltd., Shenzhen, China) has introduced a new chemiluminescence immunoassay (CLIA) for the detection of the cTn complex. The present study aims to verify and validate the hs-cTnI Mindray assay on the new automated CL2600i analyzer compared to the routine Alinity-i series instrument by Abbott (Abbott, Chicago, IL, USA). **Methods**: This study evaluated linearity, precision through the 5 × 5 protocol, methodological comparison on plasma and serum matrices, hs-cTnI 99th percentile imprecision, and the hs-cTnI detection rate in a healthy population. **Results**: The results obtained proved that the performance of the Mindray hs-cTnI test on the CL2600i platform was closely comparable to the Abbott Alinity-i system (plasma R^2^: 0.974; serum R^2^: 0.995). The CVs were consistently low, and no significant differences were reported. Excellent analytical performance, with high sensitivity, was also observed in the healthy population (overall detection rate: 79%), as well as good linearity within the measuring range (R^2^: 0.994). **Conclusions**: The Mindray hs-cTnI test confirms its robustness and utility in routine practice as an advanced assay. The new technology, with more sensitive detection methods, may improve the accuracy and reliability of cardiac biomarker testing, ultimately leading to better outcomes in the management of patients with AMI and other cardiac conditions.

## 1. Introduction

Cardiac troponin, particularly high-sensitivity cardiac troponin (hs-cTn) I and T tests are essential tools for the early and accurate diagnosis of acute myocardial infarction (AMI) [1]. In 2018, the Fourth Universal Definition of Myocardial Infarction [2] established that a myocardial injury is present when a single hs-cTnI or hs-cTnT value above the 99th percentile URL (Upper Reference Limit) occurs. Accordingly, the most recent international guidelines recommend the use of hs-cTnI and hs-cTnT methods for the detection of myocardial injury as a pre-requisite for the diagnosis of AMI in patients admitted to the emergency department [2,3]. In patients undergoing cardiac revascularization, particularly per cutaneous procedures, measuring hs-cTn levels is also essential for identifying any per procedural ischemic episodes. As recently shown, these occurrences are not unusual and significantly affect prognosis [4].

Moreover, the 2018 Expert opinion from the International Federation Clinical Chemistry and Laboratory Medicine (IFCC) and American Association for Clinical Chemistry (AACC) [5] recommends that the high-sensitivity assay for cTnI and cTnT should meet two key criteria: 1. it should measure the 99th percentile URL with an imprecision (expressed as Coefficient of Variation, %CV ≤ 10; 2. it should measure cTn concentrations above the Limit of Detection (LoD) of the method in at least 50% of healthy subjects enrolled in large populations, including more than 300 men and 300 women [6].

Circulating levels of hs-cTnI and hs-cTnT have been found to increase progressively after 55 years of age in asymptomatic men and women enrolled in multicenter studies including large reference populations [3,7,8,9]. Many clinical studies and some meta-analyses have recently confirmed that some individuals, apparently free from any cardiac disease, have hs-cTnI or hs-cTnT concentrations in the third tertile of biomarker distribution values (i.e., still below the 99th percentile URL cut-off value), presenting a higher risk of early cardiac or non-cardiac mortality and rapid progression to heart failure [10,11]. Additionally, it is important to emphasize that hs-cTnI and hs-cTnT are distinct cardiac-specific biomarkers, each exhibiting a biological variability around 10%CV. The individuality index expresses the relationship between the intra-individual variability and the general variability among individuals for a specific biomarker. For cardiac troponins, the individuality index is ≤0.3, making repeated single measurements more clinically manageable compared to the analytical range [12]. Considering the methodological point of view, other manufacturers have developed different immunoassays to measure cTn in serum and/or plasma. In this context, it is possible to understand differences registered among different commercial immunoassays with regard to reference ranges, specimen type (serum, heparin/EDTA plasma), different epitope specificity of capture and detection antibodies, LoD, and percentage of %CV at the 99th percentile [13].

Furthermore, the IFCC Task Force on Clinical Applications of Cardiac Biomarkers addressed the question of whether cTn harmonization could be achieved through a secondary reference material based on a pooled serum. However, harmonization between assays remains elusive [14]. Nowadays, the market is addressing cTnI, as an assay target, rather than cTnT. The cTnI appears to be a more specific cardiac biomarker of AMI, whereas cTnT is more strongly associated with the risk of non-cardiovascular disease [15]. However, measurements of cTnI are influenced by multiple factors, including proteolytic degradation, phosphorylation, complexing with cTnC, heparin, heterophile or human antimouse antibodies, and cTnI-specific autoantibodies [16]. In addition, the path of hs-cTns has become even more uncertain with the renewed interest in hs-cTns fragments [17,18,19].

Recently, Mindray (Mindray Bio-Medical Electronics Co., Ltd., Shenzhen, China) has introduced a new chemiluminescence immunoassay (CLIA) for hs-cTnI tests. An innovative method with a detection antibody directed towards cTn complexes is used. The most novel feature of this assay is the cTn-complex-assay target. Several studies indicated that cTnI mainly exists in the forms of binary cTnIC complexes and ternary cTnITC complexes in circulation with little free cTnI [20,21]. The Mindray hs-cTnI assay utilized anti-cTn complex antibodies for the detection of the total concentration of binary cTnIC complexes and ternary cTnITC complexes, which was theoretically equivalent to the cTnI concentration. Two papers have been recently published about the verification/validation of hs-cTnI assay. Li et al. validated the Mindray CL8000i instrument in comparison with the Abbott Architect-I [22], while Lippi et al. compared the Mindray CL1200i with Access hs-TnI Beckman Coulter [23]. Results of these studies both demonstrated that the overall performance of the new Mindray assay is comparable to other commercially available methods.

In this study, we report our experience in verifying and validating the Mindray c-hsTnI assay on CL2600i in comparison with the Abbott Alinity-i.

## 2. Materials and Methods

### 2.1. Study Design

Serum or plasma samples were collected from residual routine specimens of patients admitted to the emergency department with chest pain symptoms or acute coronary syndrome. A total of 49 lithium heparin plasma and 37 serum samples were used for the comparison of cTnI assay between Alinity-i (Abbott, Abbott Park, IL, USA) and CL-2600i (Mindray, Shenzhen, China) instruments. Samples were centrifuged at 2000× *g* for 10 min and then processed according to the manufacturer’s instructions.

A total of 400 leftover samples (200 from males and 200 from females, age 18–60 years old, Caucasian ethnicity) from blood donors were collected as the healthy population to evaluate the 99th percentile. The inclusion criteria for the healthy population’s samples were glomerular filtration rate, glycosylated hemoglobin, and NT-proBNP in the range of normality. Participants had no symptoms of heart disease or any history of cardiovascular events. The exclusion criteria were neoplastic diseases, chronic and acute kidney disease, malignancies, recent infections or pharmacological treatment.

This retrospective study was evaluated and approved by the local Ethics Committee at Tor Vergata University Hospital (ID number 41.17) and was carried out according to the Declaration of Helsinki, as revised in 2013.

### 2.2. Chemiluminescence Immunoassays

The CL-2600i is a new automated immunochemistry (Mindray Bio-Medical Electronics Co., Shenzhen, China) instrument based on the “sandwich” principle. The innovation of the method includes a 4-step magnetic separation system to ensure the effective detachment of immune complexes from magnetic particles. The sample is incubated with the pretreatment solution, paramagnetic microparticles with mouse monoclonal anti-cTnI antibody (HyTest), and mouse monoclonal anticTnI-complex antibodies (HyTest) conjugated with alkaline phosphatase (ALP). After washing, addition of the chemiluminescent substrate (3-(2-helical adamantane) 4-methoxy-4-(3-phosphoxyyl)-phenyl-1,2-dioxane, AMPPD) produces chemiluminescence. The photon counting technique, using a photomultiplier (PMT), detects the light signals produced by the chemiluminescence reaction, converting the signal into a concentration (ng/L). The device is calibrated via an LED reference module, ensuring high measurement accuracy.

The internal quality controls (iQC) were analyzed at every work session.

### 2.3. Precision Evaluation

Precision estimations were obtained using quintuplicate measurements of aliquots of three pools (low, high, and very high levels of cTnI), performed for a total of 5 replicates each day, respectively, for a total of five days, following the Clinical and Laboratory Standards Institute (CLSI) EP15-A3 protocol [24]. Results obtained for precision evaluation were compared to those claimed by the manufacturer.

### 2.4. Linearity Assessment

Samples at eight different concentrations within the expected range were used to verify linearity. High-concentration samples were created by pooling samples at a concentration near the limit of linearity; they were then diluted with negative samples, as shown in Table 1. The test was performed three times for each sample concentration. The mean value and standard deviation were then calculated, and the least squares method was used to fit a straight line between the resulting mean values and the dilution ratios. Finally, the correlation coefficient r was calculated in the linear range.

### 2.5. Repeatability 

The repeatability was assessed, analyzing quality controls for each level 20 times. The results are expressed as %CVs, Standard Deviation (SD), mean. The acceptance criteria were set at %CV ≤ 5.

### 2.6. Methodological Comparison

The comparability of the Mindray hs-cTnI methodological test performed on the CL-2600i platform was assessed with the Abbott hs-cTnI assay on the Alinity-i series. The same samples processed on the Abbott Alinity-i series were subsequently measured on the CL-2600i instrument within three hours.

### 2.7. hs-cTnI 99th Percentile Imprecision

To calculate the hs-cTnI 99th percentile imprecision, the EP15-A3 protocol was followed [24]. The imprecision profile was plotted according to cTnI concentrations measured with corresponding CVs observed in a series of 30 samples, with hs-cTnI concentrations ranging between LoD values and 99th percentile URLs.

### 2.8. hs-cTnI Detection Rate in the Healthy Population

The 99th URLs were determined using 400 serum specimens from the healthy population. Based on the measurements of serum specimens obtained by the laboratory records, the sex-specific 99th-percentile URLs were calculated. The detection rate in a healthy population was calculated according to the following formula: number of samples above LoD/total number of samples (LoD values of 0.5 ng/L). The detection rate for the male, female, and overall healthy populations should be ≥50%.

### 2.9. Statistical Analysis

The 99th percentile upper reference limits (URLs) were determined using serum specimens from a healthy population as previously described. The coefficient of variation (%CV) was calculated as the standard deviation divided by the mean value. Data were considered using standard deviation (SD), Spearman rank correlation coefficient (R^2^), and linear regression equations. Passing–Bablok regression analysis was used to compare the two instruments (Mindray vs. Abbott) and the two matrices (serum vs. plasma and plasma vs. plasma), while the Bland–Altman test was performed to determine the agreement and to investigate the true value (i.e., proportional bias).

All data were analyzed using Med Calc Ver.18.2.18 (MedCalc Software Ltd., Ostend, Belgium) and OriginLab Ver. 6.1 (Northampton, MA, USA).

## 3. Results

### 3.1. Precision Study

Precision data of the coefficients of variation obtained in the laboratory are shown in Table 2 in comparison with the data declared by the manufacturer. No significant difference was observed between the %CVs of the two methods (*p* > 0.05: n.s.), even though the CV values evaluated in the laboratory were slightly higher than the %CV values declared.

### 3.2. Linearity Analysis

The linearity measurements performed on the Mindray CL-2600i instrument are shown in Figure 1. The dilution performed was obtained by diluting high-value pools with low-value pools. All tested pools of serum samples do not deviate from linearity, with an equation of y = 25,518.56 − 3194.05x.

### 3.3. Repeatability

Repeatability was assessed by analyzing the three iQC levels 20 times. Results were reported in Table 3. The %CVs evaluated met the acceptance criteria and were CV < 5%.

### 3.4. Methodological Comparison

Two linear regressions were built using Passing–Bablok analysis to compare the two methods with two different matrices: plasma vs. plasma and plasma vs. serum.

A significant correlation between the two methods for plasma vs. plasma (*n* = 49) was observed (Figure 2A), with a correlation coefficient of 0.974 (95% Confidence Interval, CI: 0.954 to 0.985) and a *p*-value less than 0.0001, indicating a strong and statistically significant correlation. The Bland–Altman analysis (Figure 2B) revealed a mean bias between the Alinity-i and CL-2600i instruments of 175.8 ng/L with the upper and lower limits of 5884.3 ng/L and −5532.7 ng/L, respectively. The probability associated with the two-sided test was non-significant (*p* = non-significant, n.s.), indicating that the differences between the methods are not statistically significant.

A significant correlation between the two methods for plasma vs. serum (*n* = 37) was observed (Figure 2C), with a correlation coefficient of 0.990 (95% CI: 0.981 to 0.995) and a *p*-value less than 0.0001, indicating again a strong and statistically significant correlation even in different matrices. The Bland–Altman analysis (Figure 2D) revealed a mean difference between the two instruments of −265.5 ng/L with the upper and lower limits of 3622.4 ng/L and −4152.2 ng/L, respectively. The probability associated with the two-sided test was non-significant (*p* = n.s.), indicating that the differences between the methods are not statistically significant.

### 3.5. hs-cTnI 99th Percentile Imprecision

The imprecision profile was plotted (Figure 3) using hs-cTnI concentrations along with the corresponding CVs and SDs observed in a series of 27 samples, with hs-cTnI concentrations ranging from LoD values to the 99th percentile URLs. Results showed a R^2^ of 0.92 and an equation of 14.52 × e^(−x/0.83)^.

### 3.6. hs-cTnI Detection Rate in Healthy Population

The 99th percentile URLs were evaluated in 200 healthy female and 200 healthy male patients.

A significant difference was observed between the cTnI values of females and males, being 1.17 ng/L and 2.28 ng/L, respectively (*p* < 0.001). Overall, the samples showed a mean of 1.72 ng/L (SD 1.44 ng/L and 95% CI from 1.52 to 1.92 ng/L).

Therefore, the detection rate in a healthy population, i.e., the number of samples above the LoD of the method on the total samples was 60% for females, 97% for males, and 79% for overall, proving the optimal performance of the test, as can be seen in Table 4.

## 4. Discussion

As technological advancements continue to enhance the precision of biomarker detection, evaluating the performance of new hs-cTn assays becomes essential. A current issue lies in the variability observed across different commercial immunoassays [25]. These discrepancies arise from various factors, including differences in the reference ranges established by various tests, the type of biological sample used (serum, heparinized plasma, or EDTA plasma), and the specificity of capture and detection antibodies for specific cardiac troponin epitopes [26].

In this context, the Mindray hs-cTnI test on the CL-2600i instrument, utilizing multiple antibodies that recognize cTnI complexes, demonstrates robust analytical performance and minimizes interferences [23], as shown in the results. This design avoided the influence of cTnI degradation and autoantibody interference and improved sensitivity and specificity, leading to better LoB, LoD, and LoQ values and anti-interference performance.

These features not only improve diagnostic accuracy, raising optimal %CVs (3.17, 2.37, and 2.02 for each level of iQC), but facilitate the integration of this method into clinical practice, showing a strong correlation with the routine method in both plasma and serum matrices (R^2^ 0.974 and R^2^ 0.995, respectively). Additionally, to lead to standardization and harmonization of results, the variability in the %CVs at the 99th percentile [21] was evaluated as a 79% detection rate for the total population, 60% for females, and 97% for males, much better than the IFCC recommendations (>50%). These data confirmed that the new Mindray assay is more sensitive than other commercially available tests [23], allowing the detection of very low cTnI concentrations in healthy populations. The use of these advanced methodologies ensures the detection of binary and ternary complexes of cTnI, capturing a broader spectrum of clinically relevant troponin forms released during myocardial injury. Indeed, a recent study explored thresholds and algorithms to investigate the best combination of cTn forms detected by specific immunoassays for diagnosing, monitoring, and follow-up of AMI [17]. This strategy, which involves developing immunoassays that detect troponin complexes rather than individual epitopes, enhances specificity and reduces the likelihood of interference from troponin degradation products or autoantibodies that could compromise assay accuracy [27] and may also be detected in non-AMI patients. High levels of cTn complex are often detected in patients with chronic disease, inflammation status, and in professional athletes [17,28,29,30]. Moreover, in this context, an assay, such as the Mindray test, that meets the criteria of standardization and high sensitivity over the 99th percentile is quite needed in routine practice. The LOD < 0.5 ng/L discriminates the minimal cTn unbalance, even in the possibly healthy individual or in patients with the slightest cardiovascular symptoms. Moreover, highly sensitive cTn can function not only as an effective diagnostic tool for cardiovascular disease (CD), but also as a marker for screening and primary prevention, predicting future risk of cardiovascular events [31,32]. Also, it could increase interest in studying the role of hs-cTnI even in other populations and ethnicities [33,34].

These results highlight the cTnI assay’s high reproducibility, particularly within clinically significant concentration ranges. Linear measurements further validated the assay performance. This indicates that the test provides accurate and proportional responses to varying troponin concentrations, ensuring reliable, even stricter quantification in diverse clinical scenarios, such as the emergency department, and avoiding misclassification [5,13,35,36].

The linearity and precision of the Mindray hs-cTnI test underline its robustness and suitability for clinical application, particularly in contexts requiring high sensitivity and reproducibility. Specifically, the %CVs were consistently low, indicating high reproducibility and reliability across a wide range of troponin concentrations.

While the Mindray hs-cTnI test (and similar tests) is designed to minimize interferences through various mechanisms like reagent optimization or advanced calibration techniques, some degree of interference is still possible, which could impact the accuracy or reliability of the results, especially in samples and autoantibodies, or with heterologous antibodies. Therefore, it is important for healthcare providers to be aware of these potential interferences when interpreting test results [34], especially in patients with autoimmune conditions or those on certain biopharmaceutical treatments.

The findings of this study support the integration of the Mindray hs-cTnI test into clinical practice, particularly in environments where rapid clinical decision-making and high accuracy are critical. Future studies should focus on the clinical application of the Mindray hs-cTnI test in larger and more diverse patient populations to confirm its robustness and utility in routine practice and the biological variability of cTnI analysis.

## 5. Conclusions

Ongoing research into the standardization and harmonization of hs-cTn assays is crucial, as variations among instruments and tests continue to pose challenges in clinical diagnostics. Advances in assay technology, including the development of even more sensitive detection methods and enhanced specificity, have the potential to improve the accuracy and reliability of cardiac biomarker tests, ultimately leading to better patient outcomes in the management of AMI and other cardiac conditions.

## Figures and Tables

**Figure 1 diagnostics-15-01031-f001:**
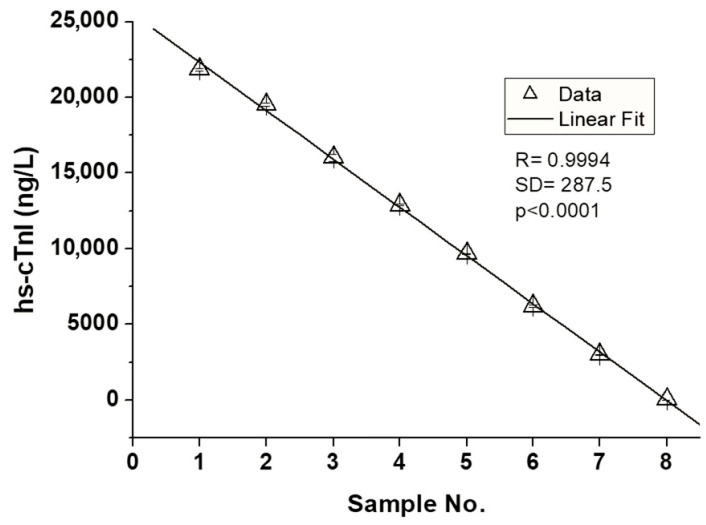
The Mindray hs-cTnI assay linearity, showing a correlation coefficient R = 0.9994. [hs-cTnI, high-sensitivity troponin I; R, correlation coefficient; SD, standard deviation].

**Figure 2 diagnostics-15-01031-f002:**
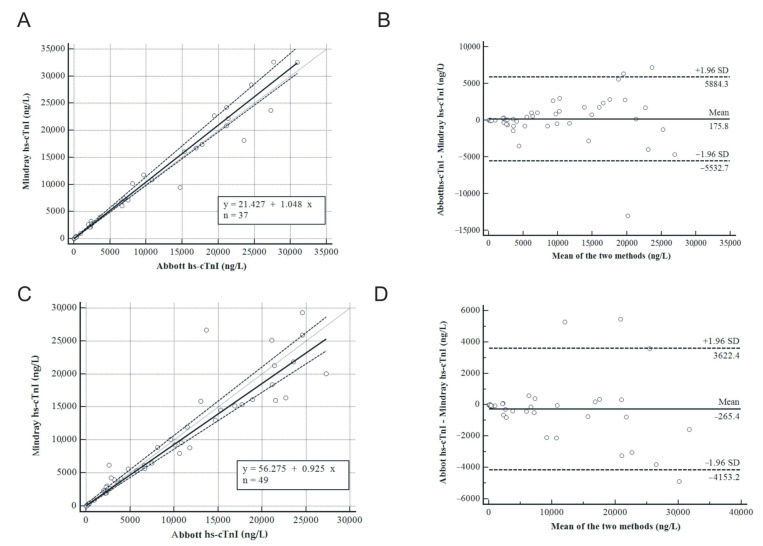
The Passing–Bablok analysis of plasma–plasma matrices (*n* = 37) between the Abbott and Mindray assays with a regression line of y= 21.427 + 1.048x (**A**) and Bland–Altman (**B**) with a mean bias of 175.8 ng/L demonstrated a strong correlation and agreement between the methods, although Mindray hs-cTnI tends to slightly overestimate values exceeding 15,000 ng/L. Passing–Bablok (**C**) analysis of plasma–serum matrices (*n* = 49) between the Abbott and Mindray assays with a regression line of y = 56.275 + 0.925x and Bland–Altman (**D**) with a mean bias of −265.5 showed a great correlation between the methods, while a non-significance increase in the distribution is observed for higher values. In (**A**,**C**) the bold line is the Passing-Bablok fit, the dotted lines are the 95%CI bands, and the grey one is the identity line. In (**B**,**D**) the bold line is the mean, the dotted lines are the upper and lower levels.

**Figure 3 diagnostics-15-01031-f003:**
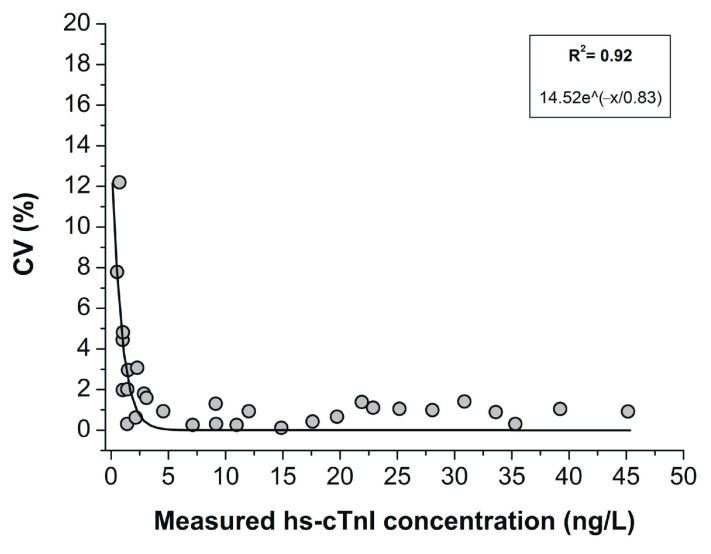
Imprecision profile using CVs and SDs for hs-cTnI, highlighting the minimal fluctuations of the hs-cTnI in values to the 99th percentile URLs.

**Table 1 diagnostics-15-01031-t001:** Concentrations for low-value samples and high-value samples for linearity study [SD: Standard Deviation].

Sample ID	Low-Value Sample	High-Value Sample	Mean ± SD (ng/L)
1	0	7/7	21,842.7 ± 66.03
2	1/7	6/7	19,539.3 ± 111.86
3	2/7	5/7	16,018.4 ± 255.53
4	3/7	4/7	12,898.2 ± 78.27
5	4/7	3/7	9665.2 ± 43.95
6	5/7	2/7	6209.5 ± 73.18
7	6/7	1/7	2986.9 ± 24.05
8	7/7	0	2.67 ± 0.23

**Table 2 diagnostics-15-01031-t002:** Precision evaluation through the 5 × 5 protocol, analyzing three iQC levels.

-	-	-	Within Run		Between Run	
TEST	Sample Pools	Mean Value (ng/L)	%CV Declared	%CV Lab	%CV Declared	%CV Lab
hs-cTnI	1	10.3	2.01	2.05	2.97	3.17
hs-cTnI	2	34.9	1.84	1.27	2.49	2.37
hs-cTnI	3	10,137.37	1.14	1.91	1.97	2.02

**Table 3 diagnostics-15-01031-t003:** Repeatability study on three iQC levels. The results are all above the 5%CV.

-	iQC 1	iQC 2	iQC 3
Mean (ng/L)	10.12	36.00	10,006.01
SD	0.42	0.79	348.44
%CV	4.19	2.19	3.48

**Table 4 diagnostics-15-01031-t004:** Evaluation of hs-cTnI detection rate in healthy population, reporting mean, standard deviation (SD), Confidence Interval, number of samples above the LoD (0.5 ng/L) and the detection rate, divided for gender.

-	Female	Male	Overall
Mean (ng/L)	1.17	2.34	1.75
SD (ng/L)	1.25	1.38	1.44
95% Confidence Interval (ng/L)	0.99–1.34	2.14–2.53	1.55–1.95
N of samples with cTnI > LoD	155 out of 200	200 out of 200	355 out of 400
Detection rate in a healthy population	78%	100%	89%

## Data Availability

Data are available under specific request to the corresponding author.

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
