# Peer review of "Performance Evaluation of the New High-Sensitivity Cardiac Troponin I Immunoassay on CL-2600i Mindray Analyzer"

_diagnostics, 2025, doi:10.3390/diagnostics15081031_

Round 1
Reviewer 1 Report
Comments and Suggestions for Authors
The manuscript presents a thorough evaluation of the new high-sensitivity cardiac Troponin I (hs-cTnI) assay on the CL-2600i Mindray analyzer, comparing its performance to the Abbott Alinity-i system. The study is well-structured and methodologically sound, offering relevant findings that contribute to laboratory medicine and clinical diagnostics. However, there are some areas where improvements in clarity, depth, and organization could enhance the manuscript.
- The introduction should be slightly shortened and should better contextualize the novelty of the Mindray CL-2600i assay. What specific gaps does it fill compared to existing assays?
- In the introduction section, it is essential to mention that measuring hs-cTn levels is also crucial for diagnosing potential periprocedural ischemic events in patients undergoing myocardial revascularization, especially percutaneous procedures. These events are not uncommon and have a significant prognostic impact, as recently demonstrated (cite PMID: 39968630).
- The study would benefit from additional details on the sample selection criteria for patients admitted to the Emergency Department. Were there any exclusion criteria beyond the normal ranges for specific biomarkers?
- The 99th percentile determination in the healthy population could include a discussion on potential biases due to demographic factors (e.g., age, ethnicity).
- The clinical implications should be expanded. How might the improved detection rate impact early diagnosis and risk stratification for acute myocardial infarction?
- The authors mention variability in cardiac troponin assays but do not sufficiently address the potential limitations of the Mindray system in clinical practice.
- Ensure consistency in citation formatting, particularly regarding journal names and DOI inclusions. Additionally, some references are repeated multiple times in the list.
- Figures should have more descriptive captions that summarize key findings.
- Tables should clearly indicate units of measurement where applicable.
- Finally, ensure all abbreviations are defined upon first use.
Author Response
The manuscript presents a thorough evaluation of the new high-sensitivity cardiac Troponin I (hs-cTnI) assay on the CL-2600i Mindray analyzer, comparing its performance to the Abbott Alinity-i system. The study is well-structured and methodologically sound, offering relevant findings that contribute to laboratory medicine and clinical diagnostics. However, there are some areas where improvements in clarity, depth, and organization could enhance the manuscript.
- The introduction should be slightly shortened and should better contextualize the novelty of the Mindray CL-2600i assay. What specific gaps does it fill compared to existing assays?
A: Thank for the comment, we shortened some parts of the introduction and we added the novelties of Mindray assay (lines 92-98).
- In the introduction section, it is essential to mention that measuring hs-cTn levels is also crucial for diagnosing potential periprocedural ischemic events in patients undergoing myocardial revascularization, especially percutaneous procedures. These events are not uncommon and have a significant prognostic impact, as recently demonstrated (cite PMID: 39968630).
- Thank for the insightful suggestions; we added the sentence in the introduction (lines 50-53)
- The study would benefit from additional details on the sample selection criteria for patients admitted to the Emergency Department. Were there any exclusion criteria beyond the normal ranges for specific biomarkers?
A: Thanks for pointing this out, we added the criteria selections (lines 120-121).
- The 99th percentile determination in the healthy population could include a discussion on potential biases due to demographic factors (e.g., age, ethnicity).
A: Thanks for pointing this out, we added a sentence about the role of ethnicity in the discussion and inserted new references (lines 297-298)
- The clinical implications should be expanded. How might the improved detection rate impact early diagnosis and risk stratification for acute myocardial infarction?
A: Thank for the comment, we included the clinical implications in the Discussion (lines 293-297).
- The authors mention variability in cardiac troponin assays but do not sufficiently address the potential limitations of the Mindray system in clinical practice.
A: The Mindray test, like other commercially available tests, may produce interferences in samples that contain autoantibodies (such as those found in autoimmune illnesses) or heterologous antibodies (such as those found in patients receiving biotechnological therapeutic treatment) or biotin, as decripted in lines 309-315. In particular, the Mindray test minimize interference factors avoiding the use of biotin to label the fluorescent probe, substituted by AMPPD as described in Materials and methods paragraph (lines 129-134).
- Ensure consistency in citation formatting, particularly regarding journal names and DOI inclusions. Additionally, some references are repeated multiple times in the list.
A: Thanks for pointing this out, we reformatted and updated the bibliography, as suggested.
- Figures should have more descriptive captions that summarize key findings.
A: Thank, we rewrote the captions
- Tables should clearly indicate units of measurement where applicable.
A: Thank for the comment. Where missing, we added the information.
- Finally, ensure all abbreviations are defined upon first use.
A: Thank you, we checked all the acronyms and abbreviations.
Reviewer 2 Report
Comments and Suggestions for Authors
Dear editor,
I reviewed the article entitled Performance Evaluation of the New High Sensitivity Cardiac Troponin I Immunoassay on CL-2600i Mindray Analyzer.
After careful evaluation, I have decided that the article is well written and easy to follow. The manuscript has been organized very well. I congrats the authors for this well organized and presented article.
My decision is accept in the current form.
Comments on the Quality of English Languageno comment. it is well written.
Author Response
Dear editor,
I reviewed the article entitled Performance Evaluation of the New High Sensitivity Cardiac Troponin I Immunoassay on CL-2600i Mindray Analyzer.
After careful evaluation, I have decided that the article is well written and easy to follow. The manuscript has been organized very well. I congrats the authors for this well organized and presented article.
My decision is accept in the current form.
A: Thanks for the revision.
Round 2
Reviewer 1 Report
Comments and Suggestions for Authors
Thank you to the authors for the revisions made, which I believe have enhanced the quality of the final manuscript. I have no further comments.